# Research on a Rapid Image Stitching Method for Tunneling Front Based on Navigation and Positioning Information

**DOI:** 10.3390/s25103023

**Published:** 2025-05-10

**Authors:** Hongda Zhu, Sihai Zhao

**Affiliations:** 1School of Mechanical and Electrical Engineering, China University of Mining & Technology-Beijing, Beijing 100083, China; zhuhongda000000@outlook.com; 2Huadian Coal Industry Group Digital Intelligence Technology Co., Ltd., Beijing 102488, China

**Keywords:** tunneling face, navigation, positioning, image stitching

## Abstract

To address the challenges posed by significant parallax, dynamic changes in monitoring camera positions, and the need for rapid wide-field image stitching in underground coal mine tunneling faces, this paper proposes a fast image stitching method for tunneling face images based on navigation and positioning data. First, using a pixel-based calculation approach, the tunneling face scene is partitioned into the cutting section and the ground, enhancing the reliability of scene segmentation. Then, the spatial distance between the camera and the cutting plane is computed based on the tunneling machine’s navigation and positioning data, and a plane-induced homography model is employed to efficiently determine the dynamic transformation matrix of the cutting section. Finally, the Dual-Homography Warping (DHW) method is applied to achieve fast panoramic image stitching of the tunneling face. Comparative experiments with three classical stitching methods, SURF, SIFT, and BRISK, demonstrate that the proposed method reduces stitching time by 60%. Field experiments in underground environments verify that this method can generate a complete panoramic stitched image of the tunneling face, providing an unobstructed perspective beyond the machine body and cutting head to clearly observe the shovel plate and surrounding ground conditions, significantly enhancing the visibility and convenience of remote operation.

## 1. Introduction

As the front line of coal mining operations, tunneling faces present significant safety risks. The implementation of intelligent remote control systems has become crucial for enhancing operational safety [1,2]. These systems rely heavily on visual data, where images and videos serve as core perception tools, enabling remote monitoring and control in coal mines. A critical challenge lies in monitoring key areas such as the cutting cross-section and the ground near the shovel plate. Traditional camera setups, dispersed across machinery [3,4,5], only capture fragmented views, failing to provide a comprehensive visual representation of these regions. This limitation hampers intuitive remote operation and impedes the practical deployment of intelligent systems [6]. Therefore, stitching together images captured by multiple cameras, especially by merging the image of the cutting face with the ground image of the scoop (shovel) board area, enables workers to observe not only the cutting face but also, as if “looking through” the obstructions of the machine body and the cutting head, the images of the scoop board and the ground. This technology can provide powerful support for the intelligentization of roadheading (tunneling/driving) operations.

Traditional image stitching methods like SURF, SIFT, and BRISK all rely on detecting key points in an image, extracting distinctive features from them, and matching these features across images to align and stitch them together. SURF (Speeded-Up Robust Features) and SIFT (Scale-Invariant Feature Transform) both focus on detecting key points that are invariant to scale, rotation, and illumination changes, making them robust for matching features in various conditions. BRISK (Binary Robust Invariant Scalable Keypoints), on the other hand, is a faster alternative that uses binary descriptors for feature matching, offering efficient performance while maintaining robustness in feature detection. These methods all aim to find overlapping areas between images and seamlessly combine them into a single, larger image by using matched key points and geometric transformations.

Building on these classic algorithms, many research groups today have expanded their work on image stitching. Ren Wei proposed installing a multi-lens panoramic camera at the front end of a roadheader in an underground coal mine to acquire large-field-of-view and high-resolution images of the roadheading face. However, due to obstructions from the machine body and the cutting head, the camera is unable to observe the ground area near the scoop board, resulting in blind spots in the field of view [7]. To obtain complete images of both the cutting face and the ground, it is necessary to install cameras on both sides and in the middle of the roadheader for image stitching. Nevertheless, significant installation position deviations among these three cameras make it difficult to avoid large parallax issues during stitching [8]. To address the problem of large parallax image stitching, Gao et al. proposed the DHW (Dual-Homography Warping) method, which segments the scene image into front and back planes and employs separate transformation matrices for each plane, combined with weighting factors for stitching alignment [9]. However, this method relies on scenarios where the camera and scene positions are relatively fixed. During coal mine tunneling, the operational equipment carrying the camera is constantly moving, causing the distance between the front and back planes in the scene image to dynamically change, making the traditional DHW method difficult to apply. For dynamic scenes within coal mine roadways, Zhang Kailong proposed utilizing online calibration of camera extrinsic parameters based on image feature points, combined with a 3D model of the equipment, to dynamically segment the front and back planes of the image [10]. However, the actual working conditions in coal mine roadways are complex, with poor lighting conditions and severe dust interference, making it difficult to obtain sufficient and high-quality feature points, thereby reducing the reliability of plane segmentation. Zhang Xuhui et al. proposed using image enhancement, feature point matching, and optimal seamline methods to calculate the transformation matrices for the front and back planes and address misalignment between them, overcoming the impact of low lighting and heavy dust underground. However, their stitching process takes nearly one second, which is difficult to meet the dynamic demands of practical applications [11].

Currently popular stitching algorithms often face issues like slow feature point matching and limitations in fixed scene applications. However, during the tunneling process, both the scene and depth of field change in real time, which makes these traditional algorithms unsuitable. To address the specific requirements of large parallax, dynamic changes in the positions of monitoring cameras, and rapid stitching of large-field-of-view images at tunneling faces in underground coal mines, this paper proposes a rapid stitching method for tunneling face images based on navigation and positioning information. Initially, based on the known coordinates of the cutting face and the ground demarcation line within the tunnel coordinate system, their corresponding pixel positions in the camera image are calculated, and the tunneling face image is subsequently divided into two planes accordingly. Furthermore, utilizing the navigation and positioning information of the roadheader, the distance between the camera and the cutting face is derived, and a dynamic transformation matrix for the cutting face is computed based on a plane-induced homography model. Finally, combined with the DHW method, rapid stitching of the cutting face images is achieved.

## 2. Segmentation and Registration of Tunneling Face

### 2.1. Calculation of Image Dynamic Segmentation Line

The navigation and positioning device for the roadheader consists of two parts: a laser guidance device suspended from the tunnel roof at the rear end of the roadheader and a pose measurement device installed on the roadheader. The origin *O*_L_ of the tunnel coordinate system is located on the laser guidance device, with the X, Y, and Z directions of the coordinate system representing the lateral, tunneling, and height directions of the tunnel, respectively. The entire navigation and positioning device provides the heading angle, pitch angle, and roll angle of the roadheader body, as well as the coordinates of the roadheader in the tunnel coordinate system [12].

To obtain surveillance images of the tunneling face, three cameras are installed on the left and right sides at the front end of the roadheader and near the driver’s position in the middle, respectively. As shown in Figure 1, the multi-camera system covers the entire cutting face and ground area through its spatially distributed fields of view.

In an actual tunnel, the boundary between the ground and the cutting face approximates an ideal straight line, and the coordinates of the endpoints *P*_1_ and *P*_2_ of this segmentation line are known in the tunnel coordinate system *O*_L_. If we calculate the imaging pixels of *P*_1_ and *P*_2_ in the cameras, the line connecting these pixels can serve as the segmentation line between the cutting face and the ground in the image.

As shown in Figure 2, taking the middle camera as an example, a calibration board is installed at the tunnel heading, and the coordinates of each corner point on the calibration board in the tunnel coordinate system can be obtained in advance through measurement. Let the coordinate of any corner point in the tunnel coordinate system be $P_{\mathrm{tar}\_\mathrm{in}\_\mathrm{OL}}$. Meanwhile, based on the camera’s intrinsic parameters, the geometric parameters of the calibration board itself, and the pixel coordinates corresponding to the image of this corner point captured by the middle camera, its coordinate in the middle camera coordinate system can be calculated as $P_{\mathrm{tar}\_\mathrm{in}\_\mathrm{OCm}}$.(1)$$P_{\mathrm{tar}\_\mathrm{in}\_\mathrm{OCm}}={\left(R_{\mathrm{Cm}}^{O_{t}}\right)}^{-1}\left[{\left(R_{O_{t}}^{O_{L}}\right)}^{-1}\cdot \left(P_{\mathrm{tar}\_\mathrm{in}\_\mathrm{OL}}-T_{O_{t}}^{O_{L}}\right)-T_{\mathrm{Cm}}^{O_{t}}\right]$$
where $R_{O_{t}}^{O_{L}}$ and $T_{O_{t}}^{O_{L}}$ represent the rotation matrix and translation vector from the pose measurement device coordinate system *O*_t_ to the tunnel coordinate system *O*_L_, which can be obtained by inverse calculation using the heading angle, roll angle, pitch angle, and spatial position coordinates provided by the roadheader navigation and positioning system. $R_{\mathrm{Cm}}^{O_{t}}$ and $T_{\mathrm{Cm}}^{O_{t}}$ represent the rotation matrix and translation vector from the middle camera coordinate system *O*_Cm_ to the pose measurement device coordinate system *O*_t_, which can be calibrated by combining multiple corner point coordinates and Equation (1).

For the endpoints *P*_1_ and *P*_2_ of the segmentation line between the cutting face and the ground, their coordinates in the tunnel coordinate system are *P*_L_1_ and *P*_L_2_, respectively, while their coordinates *P*_Cm_1_ and *P*_Cm_2_ in the middle camera coordinate system can be obtained using Equation (2).(2)$$\left\{\begin{array}{c} P_{\mathrm{Cm}\_1}=R_{O_{t}}^{\mathrm{Cm}}\left(R_{O_{L}}^{O_{t}}\cdot P_{L\_1}+T_{O_{L}}^{O_{t}}\right)+T_{O_{t}}^{\mathrm{Cm}}\\ P_{\mathrm{Cm}\_2}=R_{O_{t}}^{\mathrm{Cm}}\left(R_{O_{L}}^{O_{t}}\cdot P_{L\_2}+T_{O_{L}}^{O_{t}}\right)+T_{O_{t}}^{\mathrm{Cm}} \end{array}\right.$$

By combining the camera’s intrinsic parameters, we can further obtain the corresponding pixel coordinates *p*_m1_ and *p*_m2_ of these two points in the image captured by the middle camera with the following equation.(3)$$\left\{\begin{array}{c} p_{m1}=\frac{1}{P_{\mathrm{Cm}1\_Z}}\cdot K_{\mathrm{Cm}}\cdot P_{\mathrm{Cm}1}\\ p_{m2}=\frac{1}{P_{\mathrm{Cm}2\_Z}}\cdot K_{\mathrm{Cm}}\cdot P_{\mathrm{Cm}2} \end{array}\right.$$
where $K_{\mathrm{Cm}}$ is the intrinsic matrix of the middle camera, while $P_{\mathrm{Cm}1\_Z}$ and $P_{\mathrm{Cm}2\_Z}$ represent the distances of points *P*_1_ and *P*_2_, respectively, along the Z-axis direction of the middle camera. The line connecting the pixel points *p*_m1_ and *p*_m2_ serves as the line between the cutting face and the ground in the tunneling heading image.

This method is also applicable to the images captured by the left and right cameras. On the one hand, it can obtain the rotation matrix $R_{\mathrm{Lm}}^{O_{t}}$ and translation vector $T_{\mathrm{Lm}}^{O_{t}}$ from the left camera coordinate system *O*_Lm_ to the pose measurement device coordinate system *O*_t_, as well as the rotation matrix $R_{\mathrm{Rm}}^{O_{t}}$ and translation vector $T_{\mathrm{Rm}}^{O_{t}}$ from the right camera coordinate system *O*_Rm_ to the pose measurement device coordinate system *O*_t_. On the other hand, it can also calculate the segmentation lines between the cutting face and the ground in the left and right camera images, as shown in Figure 2. *p*_r1_, *p*_r2_ and *p*_l1_, *p*_l2_ are the corresponding pixel points of points *P*_1_ and *P*_2_ in the right and left camera images, respectively, and their connecting lines represent the segmentation lines between the cutting face and the ground in the corresponding images.

Compared to other methods that utilize image feature points to divide the front and back planes, this method directly calculates and generates the segmentation lines between the cutting face and the ground based on the coordinates of the segmentation line endpoints, thereby avoiding reliance on image quality and significantly improving the reliability of the segmentation of the tunneling heading image.

### 2.2. Stitching of Segmented Images

After completing the segmentation of the cutting face and the ground within the front-facing images, this section employs the DHW method for these two planes. It separately transforms the images captured by the left and center cameras into the perspective of the right camera through matrix transformation, and then stitches them with the image captured by the right camera.

#### 2.2.1. Stitching of Ground Plane

As shown in Figure 1, the heights of the left, center, and right cameras remain basically unchanged relative to the roadway ground. Therefore, once the camera installation positions are fixed, the homography matrices $H_{l\_\mathrm{ground}}^{r\_\mathrm{ground}}$ and $H_{c\_\mathrm{ground}}^{r\_\mathrm{ground}}$ for perspective transformation from the left and center cameras to the right camera are also fixed. These matrices can be pre-calibrated by setting multiple feature points in the common ground area of the cameras and performing feature point matching.

#### 2.2.2. Stitching of Cutting Face

Unlike the ground, the distance between the cutting face and the cameras varies with the movement of the roadheader, leading to dynamic adjustments in the transformation matrices between images captured by different cameras according to this distance. To address the dynamically changing transformation matrices, this paper adopts a plane-induced homography model, which is rapidly calculated by integrating the internal and external parameters of the cameras, the pose transformation matrices between the coordinate systems of each camera, the normal vector of the cutting face in the camera coordinate system, and the distance from the camera to the cutting face [13].

Taking the left camera as an example, the homography matrix $H_{\mathrm{Lm}\_\mathrm{wall}}^{\mathrm{Rm}\_\mathrm{wall}}$ of the cutting face portion within its image relative to the cutting face portion in the right camera’s image can be calculated by Equation (3).(4)$$H_{\mathrm{Lm}\_\mathrm{wall}}^{\mathrm{Rm}\_\mathrm{wall}}=K_{\mathrm{Lm}}\cdot R_{\mathrm{Lm}}^{\mathrm{Rm}}\left(I+\frac{T_{\mathrm{Lm}}^{\mathrm{Rm}}\cdot n_{\mathrm{wall}}^{\mathrm{Rm}}}{d_{\mathrm{wall}}^{\mathrm{Rm}}}\right)\cdot K_{\mathrm{Rm}}^{-1}$$
where $K_{\mathrm{Lm}}$ and $K_{\mathrm{Rm}}$ are the intrinsic matrices of the left and right cameras, respectively; $R_{\mathrm{Lm}}^{\mathrm{Rm}}$ and $T_{\mathrm{Lm}}^{\mathrm{Rm}}$ are the rotation matrix and translation matrix from the left camera to the right camera, respectively; *I* is the identity matrix; $n_{\mathrm{wall}}^{\mathrm{Rm}}$ is the normal vector of the cutting face in the right camera coordinate system; and $d_{\mathrm{wall}}^{\mathrm{Rm}}$ is the distance from the cutting face to the origin of the right camera coordinate system.

Based on the coordinate system transformation relationship, $R_{\mathrm{Lm}}^{\mathrm{Rm}}$ and $T_{\mathrm{Lm}}^{\mathrm{Rm}}$ can be obtained according to the positional relationships between the left and right camera coordinate systems and the pose measurement device coordinate system provided in Section 2.1.(5)$$\left\{\begin{array}{c} R_{\mathrm{Lm}}^{\mathrm{Rm}}={\left(R_{\mathrm{Rm}}^{O_{t}}\right)}^{-1}\cdot R_{\mathrm{Lm}}^{O_{t}}\\ T_{\mathrm{Lm}}^{\mathrm{Rm}}={\left(R_{\mathrm{Rm}}^{O_{t}}\right)}^{-1}\cdot T_{\mathrm{Lm}}^{O_{t}}+T_{O_{t}}^{\mathrm{Rm}} \end{array}\right.$$

$d_{\mathrm{wall}}^{\mathrm{Rm}}$ and $n_{\mathrm{wall}}^{\mathrm{Rm}}$ can be determined based on several feature points (such as *P*_1_, *P*_2_, etc.) on the cutting face and the coordinate values of the right camera in the roadway coordinate system *O*_L_ [13,14]. The specific calculation steps are as follows: As shown in Figure 2, four feature points *P*_1_, *P*_2_, *P*_3_, and *P*_4_ are selected on the cutting face (*P*_1_ and *P*_2_ are the endpoints of the intersection line between the cutting face and the ground, and *P*_3_ and *P*_4_ are two feature points on the cutting face directly above *P*_1_ and *P*_2_ at a height *h*). Their coordinates in the roadway coordinate system *O*_L_ are known, denoted as *P*_L_1_, *P*_L_2_, *P*_L_3_, and *P*_L_4_, respectively. By combining the rotation matrix $R_{O_{t}}^{Rm}$ and translation vector $T_{O_{t}}^{Rm}$ between the pose measurement device and the right camera obtained through calibration in Section 2.1, as well as the pose parameters provided by the navigation and positioning system, their coordinates in the right camera coordinate system can be calculated.(6)$$\left\{\begin{array}{c} P_{Rm\_1}(x_{1},y_{1},z_{1})=R_{O_{t}}^{Rm}\left[R_{O_{L}}^{O_{t}}\left(P_{L\_1}+T_{O_{L}}^{O_{t}}\right)+T_{O_{t}}^{Rm}\right]\\ P_{Rm\_2}(x_{2},y_{2},z_{2})=R_{O_{t}}^{Rm}\left[R_{O_{L}}^{O_{t}}\left(P_{L\_2}+T_{O_{L}}^{O_{t}}\right)+T_{O_{t}}^{Rm}\right]\\ P_{Rm\_3}(x_{3},y_{3},z_{3})=R_{O_{t}}^{Rm}\left[R_{O_{L}}^{O_{t}}\left(P_{L\_3}+T_{O_{L}}^{O_{t}}\right)+T_{O_{t}}^{Rm}\right]\\ P_{Rm\_4}(x_{4},y_{4},z_{4})=R_{O_{t}}^{Rm}\left[R_{O_{L}}^{O_{t}}\left(P_{L\_4}+T_{O_{L}}^{O_{t}}\right)+T_{O_{t}}^{Rm}\right] \end{array}\right.$$

Subsequently, the normal vector $n_{\mathrm{wall}}^{\mathrm{Rm}}$ of the cutting face formed by the four points *P*_1_, *P*_2_, *P*_3_, and *P*_4_ in the right camera coordinate system and the distance $d_{\mathrm{wall}}^{\mathrm{Rm}}$ from the cutting face to the origin of the right camera coordinate system can be obtained.(7)$$n_{\mathrm{wall}}^{\mathrm{Rm}}\left(x,y,z\right)=\left\{\begin{array}{c} x\left(x_{1}-x_{2}\right)+y(y_{1}-y_{2})+z(z_{1}-z_{2})=0\\ x\left(x_{1}-x_{3}\right)+y(y_{1}-y_{3})+z(z_{1}-z_{3})=0\\ x\left(x_{1}-x_{4}\right)+y(y_{1}-y_{4})+z(z_{1}-z_{4})=0 \end{array}\right.$$(8)$$d_{\mathrm{wall}}^{\mathrm{Rm}}=\frac{\left|\left(O_{\mathrm{Rm}}-P_{\mathrm{rm}\_1}\right)\cdot n_{\mathrm{wall}}^{\mathrm{Rm}}\right|}{\left|n_{\mathrm{wall}}^{\mathrm{Rm}}\right|}$$

By substituting the results from Equations (5)–(8) into Equation (4), the dynamic homography matrix $H_{\mathrm{Lm}\_\mathrm{wall}}^{\mathrm{Rm}\_\mathrm{wall}}$ of the left camera relative to the right camera can be calculated. Similarly, the homography matrix $H_{\mathrm{Cm}\_\mathrm{wall}}^{\mathrm{Rm}\_\mathrm{wall}}$ of the cutting face portion within the middle camera image relative to the right camera can also be determined.

The aforementioned process demonstrates that, given the camera parameters, the proposed method in this paper can quickly solve for the transformation matrix of the cutting face by combining the coordinate information of the cutting face provided by the navigation and positioning system and the distance from the camera to the cutting face. This approach avoids the complex process in traditional algorithms of first selecting feature points, performing registration, and then calculating the homography matrix through SVD decomposition, thereby significantly improving real-time performance.

#### 2.2.3. Stitching of the Overall Image

After completing image segmentation and the calculation of transformation matrices, when employing the DHW method for final stitching, it is necessary to calculate the weight of each pixel based on its position in the image and perform overlay fusion. The process is illustrated in Figure 3.

Assuming there exists a pixel point *p* in the left image, the transformation matrix *H_p_* used to convert it to the right camera’s perspective can be calculated according to the following formula.(9)$$H_{p}=\left(1-\omega_{p}\right)\cdot H_{\mathrm{Lm}\_\mathrm{wall}}^{\mathrm{Rm}\_\mathrm{wall}}+\omega_{p}\cdot H_{\mathrm{Lm}\_\mathrm{ground}}^{\mathrm{Rm}\_\mathrm{ground}}$$
where $\omega_{p}$ is the weight corresponding to the position of point *p* in the image, and the calculation method is as follows.(10)$$\omega_{p}=\frac{L_{p\_\mathrm{wall}}}{L_{p\_\mathrm{wall}}+L_{p\_\mathrm{ground}}}$$

$L_{p\_\mathrm{wall}}$ represents the distance from pixel point *p* to the nearest feature point on the cutting face, and $L_{p\_\mathrm{ground}}$ represents the distance from pixel point *p* to the nearest feature point on the ground. Since the segmentation line between the cutting face and the ground has been clearly marked in this paper, when point *p* is on the cutting face, $L_{p\_\mathrm{wall}}=0$, and at this time, $\omega_{p}=0$; similarly, when point *p* is on the ground, $L_{p\_\mathrm{ground}}=0$, and at this time, $\omega_{p}=1$.

After performing perspective transformation on the left camera image using Equation (9), it is stitched and fused with the right camera image. The position of the image stitching seam (as indicated by the annotated stitching transition region in the figure) can achieve a smooth transition through linear weight blending [15].

After completing the stitching of the left and right camera images, this stitched image is then fused with the middle camera image. Since, in practical scenarios, the middle camera is usually installed at a higher position above the ground compared to the left and right cameras, the cutting face portion occupies a larger proportion in its image. Therefore, only the cutting face portion (above the dynamic segmentation line) in the middle camera image is selected, subjected to perspective transformation using the dynamic homography matrix $H_{\mathrm{Cm}\_\mathrm{wall}}^{\mathrm{Rm}\_\mathrm{wall}}$ and then fused and stitched a second time with the previously stitched image of the left and right cameras to ultimately obtain a complete stitched image of the three cameras, as shown in Figure 4 below.

## 3. Experimental Verification

### 3.1. Introduction to the Simulation Experimental System

To verify the stitching effectiveness and efficiency of the method proposed in this paper, a simulation experimental environment for roadways was established indoors, as shown in Figure 5 below, by referencing actual underground tunneling face scenarios.

As shown in Figure 5a, the corridor section is assumed to be a tunneling roadway, with the wall directly ahead (enclosed by the red line segment) being the cutting face. A trolley in the middle of the roadway simulates the tunneling machine, equipped with a navigation and positioning system for the tunneling machine that provides real-time location and attitude information of the tunneling machine within the roadway. With reference to actual working conditions, the laser guidance device (i.e., the origin of the roadway coordinate system *O*_L_) in the tunneling machine navigation and positioning system is located approximately 18 m behind the machine in the middle of the roadway. The coordinates of endpoints *P*_1_ and *P*_2_, which are the intersection points between the corresponding simulated cutting face and the ground plane, are known in *O*_L_.

In Figure 5b, the left and right cameras are mounted on the left and right sides of the simulated tunneling machine, respectively, with a horizontal downward viewing angle of approximately 25° towards the ground and a height of approximately 50 cm above the ground. The middle camera is located in the middle of the machine body, with a horizontal viewing angle towards the cutting face and a height of approximately 100 cm above the ground. The simulated roadway images captured by the three cameras are shown in Figure 6. We deployed pieces of A4 paper sheets on the low-textured floor surface as artificial reference markers to demonstrate the quality of the image stitching result.

### 3.2. Three-Camera Image Stitching Process

Following the method described in Section 2, the specific stitching process is as follows:

(1) Calculation of Image Segmentation Lines

First, the rotation matrices and translation vectors from the pose measurement device coordinate system *O*_t_ to each camera coordinate system are calibrated using the method in Section 2.1. Simultaneously, given the coordinates of the two endpoints *P*_1_ and *P*_2_ of the boundary line between the cutting face and the ground plane in *O*_L_, the segmentation lines between the cutting face and the ground in each camera image are calculated and displayed in the corresponding images, as shown in Figure 6.

(2) Image Fusion and Stitching of Left and Right Cameras

As described in Section 2.2, the transformation matrix $H_{\mathrm{Lm}\_\mathrm{ground}}^{\mathrm{Rm}\_\mathrm{ground}}$ from the left camera’s ground portion to the right camera’s ground portion is calculated using common feature points in the left and right camera images, such as the marked points shown in Figure 7.

Four feature points, *P*_1_, *P*_2_, *P*_3_, and *P*_4_, are selected on the cutting face. Among them, *P*_1_ and *P*_2_ are points on the segmentation line between the cutting face and the ground, while *P*_3_ and *P*_4_ are two feature points on the cutting face located 1 m directly above *P*_1_ and *P*_2_, respectively. The coordinates of these four points are substituted into Equations (6)–(9) to calculate the dynamic transformation matrix $H_{\mathrm{Lm}\_\mathrm{wall}}^{\mathrm{Rm}\_\mathrm{wall}}$ of the left camera relative to the right camera and the dynamic transformation matrix $H_{\mathrm{Cm}\_\mathrm{wall}}^{\mathrm{Rm}\_\mathrm{wall}}$ of the middle camera relative to the right camera. After applying a homography matrix weighted transformation to the left image using Equation (9), it is stitched with the right camera image. Figure 8 demonstrates the stitching effect of the left camera image transformed using either the cutting face homography matrix or the ground homography matrix individually and then stitched with the right camera image, along with a comparison to the stitching effect using the method proposed in this paper. As shown in Figure 8, if only the homography matrix corresponding to the cutting face is used, misalignment occurs in the ground part of the stitched image; if only the homography matrix corresponding to the ground is used, large-scale deformation occurs in the cutting face part of the stitched image; thus, using the segmentation and weighted transformation method proposed in this paper significantly improves the stitching effect.

(3) Three-camera image stitching

The dynamic homography matrix $H_{\mathrm{Cm}\_\mathrm{wall}}^{\mathrm{Rm}\_\mathrm{wall}}$ from the middle camera to the right camera is used to perform perspective transformation on the cutting face section, which is then stitched with the fused result of the left and right camera images as mentioned above. Ultimately, a complete stitched image from the three-camera fusion is obtained, as shown in Figure 9. It can be observed that the stitched image eliminates the obstruction of the roadheader’s own components to the line of sight, presenting a complete view of the heading face scene.

### 3.3. Time Consumption Performance Analysis

To verify the effectiveness of the proposed algorithm in terms of time efficiency, three classic stitching algorithms—SURF [16], SIFT [17], and BRISK [18]—were employed to perform full-image stitching based on the left and right camera images in Figure 8. The computational time of each method was compared (using the average time taken over 10 executions of each method), as shown in Table 1 below.

The stitching quality of the three traditional methods (SURF, SIFT, and BRISK) is comparable to the method proposed in this paper, and all of them can meet the requirements for tunneling monitoring. However, the algorithm presented in this paper has a clear advantage in terms of time consumption. Compared with classic stitching algorithms, the computational time is reduced by over 60%. Under the same hardware acceleration conditions, the proposed algorithm is more capable of meeting real-time monitoring requirements.

## 4. Verification of Real Coal Mine Tunneling Scenario

The proposed method in this paper was practically validated in a real coal mine tunneling scenario, as shown in Figure 10 below. The left and right cameras were installed on the left and right sides of the front end of the roadheader body, approximately 1 m above the ground and tilted downward at an angle of approximately 25° to monitor the scraper board and the ground in front of it. The middle camera was installed near the driver’s position, approximately 2 m above the ground, to monitor the cutting face section.

After pre-calibrating the transformation relationship between the camera and the inertial navigation system on the roadheader body, the perspective of the middle camera is used as the stitching reference. The images captured by the three cameras, as well as the segmentation lines between the cutting cross-section and the ground in each camera image, are shown in Figure 11 below.

The stitching process of images from the three cameras is illustrated in Figure 12. It can be observed that in the original, unstitched images, the field of view of the middle camera only captures partial information of the cutting cross-section, while the ground information below the cutting head is completely obscured. However, after stitching, not only can the complete image information of the cross-section be observed, but it is also possible to “see through” the cutting head to view the full scoop plate and ground information beneath it. Meanwhile, we also conducted experiments on the continuous image stitching of the underground tunneling process. The experimental results show that both the stitching quality and speed meet the monitoring requirements for the underground tunneling process. Since navigation and position information are employed, the increase in interference such as dust and low lighting does not reduce the processing speed of the algorithm presented in this paper. These features significantly enhance the visibility and operational precision of remote operations, particularly in assisting with machine relocation and clearing loose coal on the floor, holding important application value.

## 5. Conclusions

In response to the special requirements of large parallax in underground coal mine tunneling faces, dynamic changes in the positions of surveillance cameras, and rapid stitching of large-field-of-view images, this paper proposes a rapid stitching method for tunneling head images based on navigation and positioning information. This method first utilizes tunnel coordinate information to divide the tunneling head scene into the cutting cross-section and the ground through pixel calculations, enhancing the reliability of image segmentation. Subsequently, it leverages tunneling machine navigation and positioning information to calculate the spatial distance between the camera and the cutting plane and employs a plane-induced homography model to efficiently solve for the dynamic homography matrices of the cutting cross-section and the ground. Finally, the DHW method is adopted to achieve rapid stitching of tunneling head images.

By comparing this method with three classical stitching methods (SURF, SIFT, and BRISK), the results demonstrate a 60% reduction in stitching time. Application validation has been conducted under actual working conditions in coal mines, yielding complete stitched images of tunneling heads. This not only allows workers to observe the entire working face area from a unified perspective but also enables them to “see through” the machine body and cutting head to observe the scoop plate and the ground nearby, greatly facilitating remote machine movement and float coal cleanup operations by workers. This method holds high engineering application value.

## Figures and Tables

**Figure 1 sensors-25-03023-f001:**
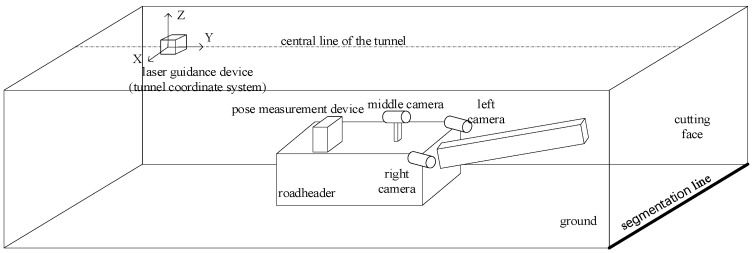
Schematic diagram of the roadheader working face and camera view.

**Figure 2 sensors-25-03023-f002:**
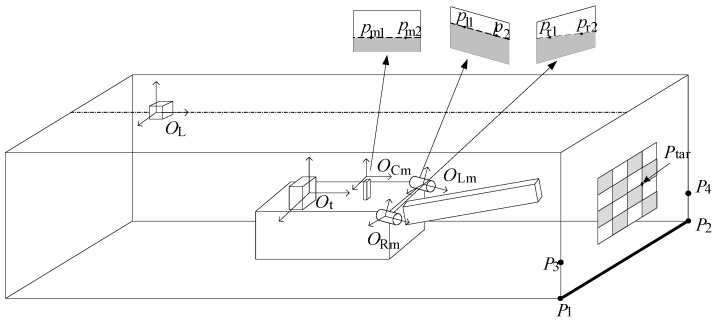
Calibration method for camera coordinate system and tunnel coordinate system.

**Figure 3 sensors-25-03023-f003:**
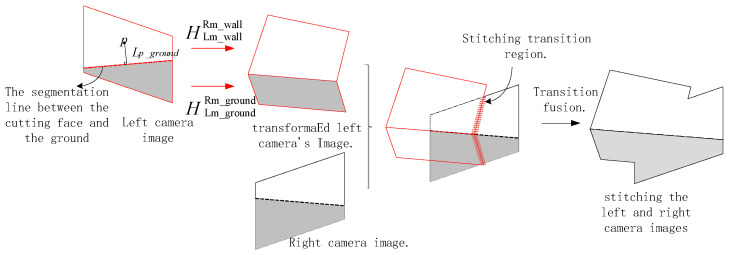
Schematic diagram of the stitching principle for left and right camera images.

**Figure 4 sensors-25-03023-f004:**
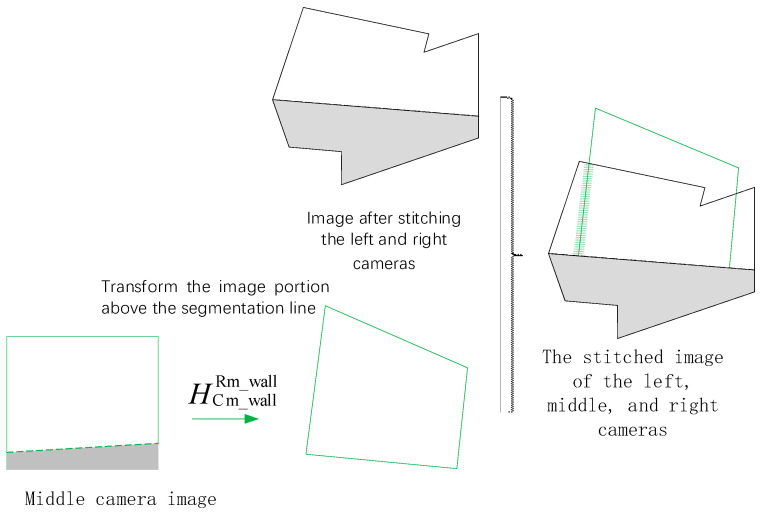
Schematic diagram of stitching for the fusion image of the middle camera with the left and right cameras.

**Figure 5 sensors-25-03023-f005:**
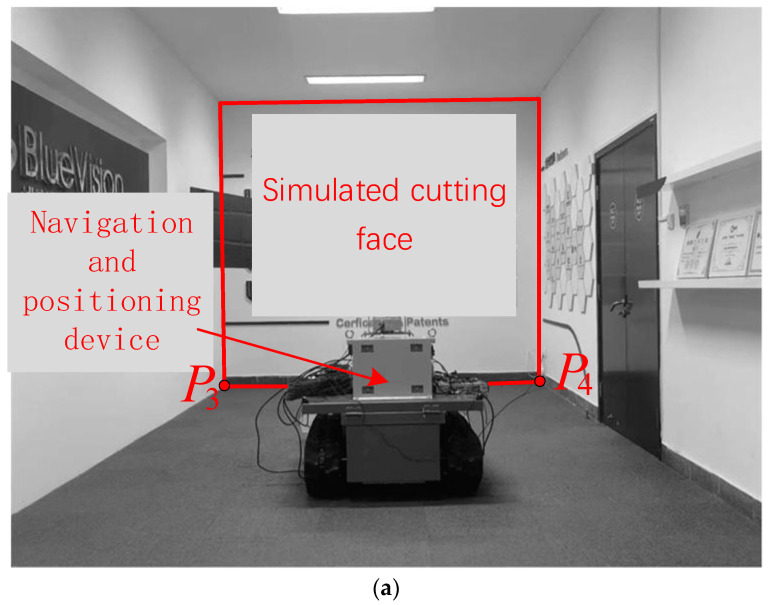
Construction of a simulation experiment system. (**a**) Schematic diagram of the simulated roadway. (**b**) Installation of simulated tunneling machine and sensors.

**Figure 6 sensors-25-03023-f006:**
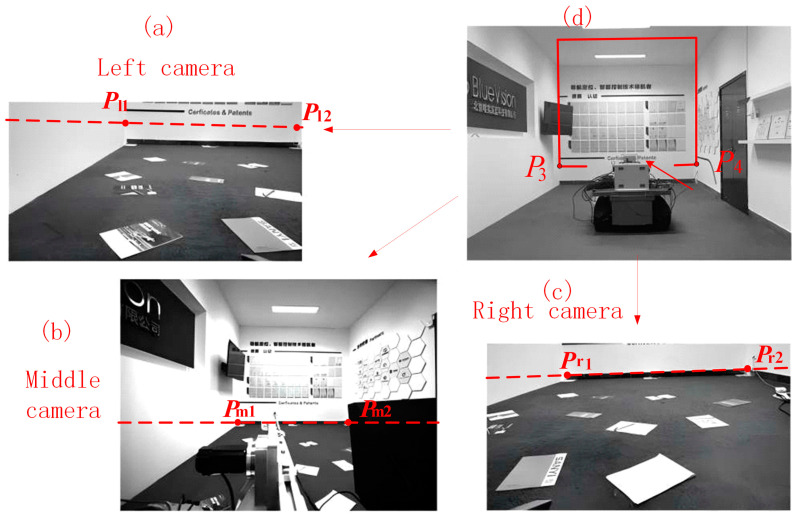
Simulated roadway images captured by the three cameras with segmentation lines marked. (**a**) The image captured by the left camera. (**b**) The image captured by the middle camera. (**c**) The image captured by the right camera. (**d**) Final stitched image.

**Figure 7 sensors-25-03023-f007:**
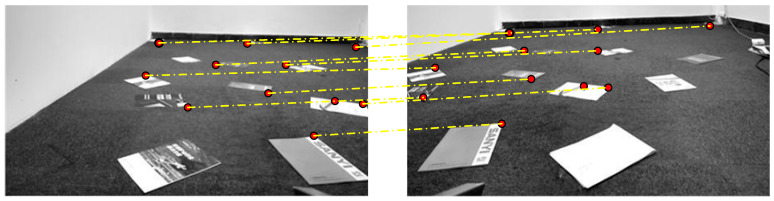
Selection of matching feature points in Figure 7’s left and right camera images.

**Figure 8 sensors-25-03023-f008:**
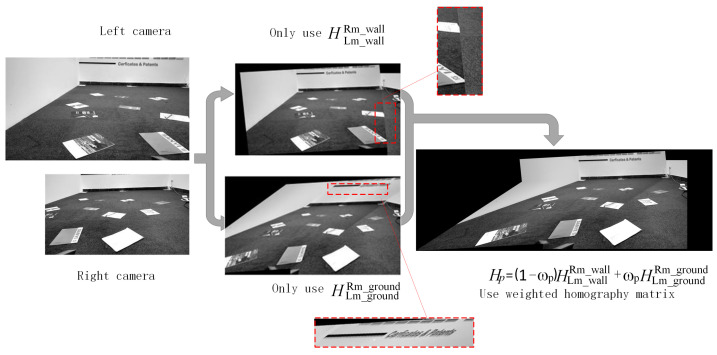
Stitching of left and right camera images.

**Figure 9 sensors-25-03023-f009:**
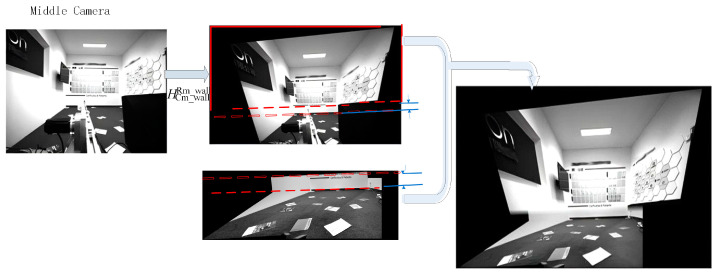
Stitching of fused images from the middle camera and the left and right cameras.

**Figure 10 sensors-25-03023-f010:**
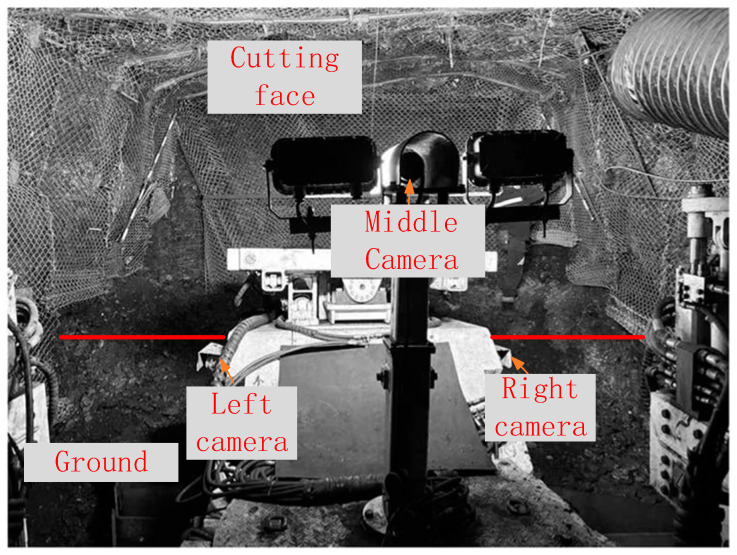
Camera position layout at the heading face.

**Figure 11 sensors-25-03023-f011:**
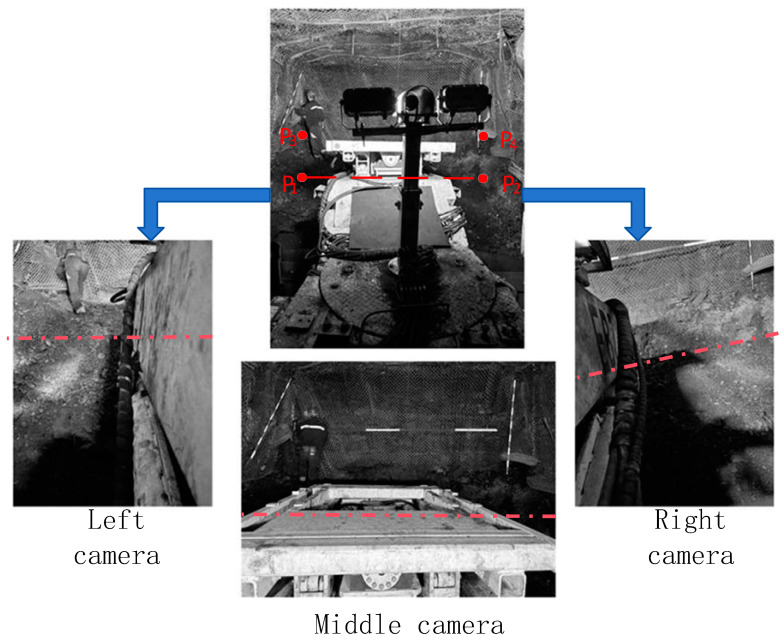
Preset feature points of the cutting cross-section and plane segmentation lines.

**Figure 12 sensors-25-03023-f012:**
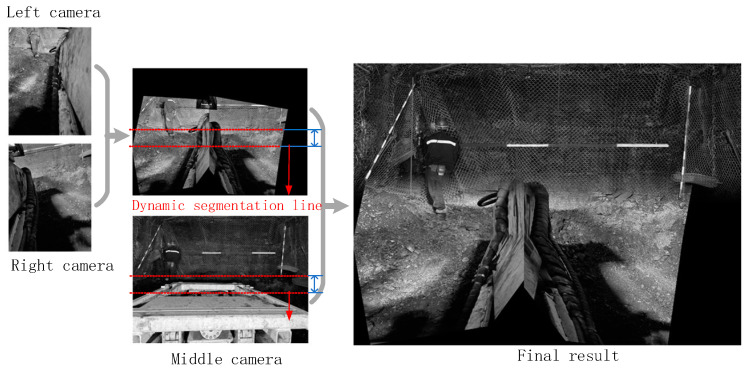
Overall stitching result in a mine.

**Table 1 sensors-25-03023-t001:** Time performance analysis of different algorithms.

Algorithm	Proposed Algorithm	SURF	SIFT	BRISK
Time (ms)	129.1	389.8	845.3	349.5

## Data Availability

Data and code of this work will be available from the corresponding author upon reasonable request.

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
