# Peer review of "Research on a Rapid Image Stitching Method for Tunneling Front Based on Navigation and Positioning Information"

_sensors, 2025, doi:10.3390/s25103023_

Round 1
Reviewer 1 Report
Comments and Suggestions for Authors
The work “Research on a Rapid Image Stitching Method for Tunneling Front Based on Navigation and Positioning Information” is devoted to the development of an efficient stitching method that meets the requirements of real-time tunneling.
The paper indicates that existing methods such as SURF, SIFT, and BRISK do not have sufficient performance, so the shortest computing time for 10 executions was 350 ms for the BRISK method. At the same time, the use of the method proposed in the work based on allows reducing this time to 129 ms. Its application requires a navigation and positioning device for the tunneling machine, including a laser guidance device and a position measuring device. This method uses the coordinate information of the tunnel obtained from the above devices to divide the tunnel head scene into a cutting cross-section and the ground using pixel calculations, and only then uses the navigation and positional information of the tunnel machine to calculate the spatial distance between the camera and the cutting plane using the homography model. Finally, the Dual-Homography Warping method is used to achieve fast stitching of tunnel head images. Compared with other considered methods SURF, SIFT, and BRISK, the proposed method reduces the stitching time by up to 60%. During field experiments in underground conditions, the approbation was carried out. The proposed method allows to create a full panoramic stitched image of the tunnel face and a view behind the machine body and the cutting head. This is necessary for assessing the position of the shovel plate and the surrounding soil conditions, which significantly improves visibility and convenience of remote control.
There are some points in the work that require corrections:
- In the description of figure 5, "blue trolley" is indicated. Since all the photographs are presented in black and white, it is not advisable to indicate its color in the description.
- In Figure 6, the summary image does not display any paper-like elements. The reason for this is not explained in the text.
- The paper states that studies were conducted in real conditions, however, no assessment of the impact of actual conditions directly during device operation in low light and strong interference from dust is given. At the same time, the paper shows that the proposed method is 60% faster than existing SURF, SIFT and BRISK. Will this speed be maintained with increasing interference?

Reviewer 2 Report
Comments and Suggestions for Authors
The work as mention is Rapid image stitching method for tunneling, in overall can proceed to next step with several comments as:
- Writing for the manuscript need to improve for example figure 6 consist of 6 image and need to elaborate similar to figure 5, as well as for figure 7 and 8.
- Tabel 1 shows the comparison results of stitching used 3 different algorithm and the best performance is used BRISK method with 349.5 ms. The results shows on how is the quality of the images after stitching need to elaborate a bit detail, its not only the response time but the quality as output the results need to consider.
- Ned to shows the testing for the multiple images and how the performance let say more than 10 images, is the output perform in good image ?
- Reference need to check, template need to clear and write the actual and some of reference does not related to work have been done.
- Presentation of image in figure 6, 8, and 9 need to enlarge a bit to get clear view.
Thank you.
